# Grape Pomace—Advances in Its Bioactivity, Health Benefits, and Food Applications

**DOI:** 10.3390/foods13040580

**Published:** 2024-02-14

**Authors:** Angélica Almanza-Oliveros, Israel Bautista-Hernández, Cecilia Castro-López, Pedro Aguilar-Zárate, Zahidd Meza-Carranco, Romeo Rojas, Mariela R. Michel, Guillermo Cristian G. Martínez-Ávila

**Affiliations:** 1Laboratorio de Química y Bioquímica, Facultad de Agronomía, Universidad Autónoma de Nuevo León, General Escobedo 66050, Mexico; aalmanzaoliveros@gmail.com (A.A.-O.); zahidd.mezaca@uanl.edu.mx (Z.M.-C.); romeo.rojasmln@uanl.edu.mx (R.R.); 2Centro de Biotecnologia e Química Fina, Laboratório Associado, Escola Superior de Biotecnologia, Universidade Católica Portuguesa, 4169-005 Porto, Portugal; s-ibhernandez@ucp.pt; 3Laboratorio de Biotecnología y Biología Molecular, Departamento de Ciencias Básicas, Universidad Autónoma Agraria Antonio Narro, Saltillo 25315, Coahuila, Mexico; c.castro.lop.28@gmail.com; 4Departamento de Ingenierías, Tecnológico Nacional de Mexico/I.T. de Ciudad Valles, San Luis Potosí 79010, Mexico; pedro.aguilar@tecvalles.mx (P.A.-Z.); mariela.michel@tecvalles.mx (M.R.M.); 5Laboratorio Nacional CONAHCYT de Apoyo a la Evaluación de Productos Bióticos (LaNAEPBi), Unidad de Servicio, Tecnológico Nacional de Mexico/I.T. de Ciudad Valles, San Luis Potosí 79010, Mexico

**Keywords:** *Vitis* spp., grape by-product, polyphenols, health benefit, food product

## Abstract

From a circular economy perspective, the appropriate management and valorization of winery wastes and by-products are crucial for sustainable development. Nowadays, grape pomace (GP) has attracted increasing interest within the food field due to its valuable content, comprising nutritional and bioactive compounds (e.g., polyphenols, organic and fatty acids, vitamins, etc.). Particularly, GP polyphenols have been recognized as exhibiting technological and health-promoting effects in different food and biological systems. Hence, GP valorization is a step toward offering new functional foods and contributing to solving waste management problems in the wine industry. On this basis, the use of GP as a food additive/ingredient in the development of novel products with technological and functional advantages has recently been proposed. In this review, we summarize the current knowledge on the bioactivity and health-promoting effects of polyphenolic-rich extracts from GP samples. Advances in GP incorporation into food formulations (enhancement of physicochemical, sensory, and nutritional quality) and information supporting the intellectual property related to GP potential applications in the food industry are also discussed.

## 1. Introduction

In recent years, food waste accumulation has caused serious environmental and economic concerns. At each stage of the supply chain, food waste represents an inefficient use of resources (e.g., labor, water, energy, etc.) and requires compensation for any effort and inputs expended [1]. Consequently, efficient approaches to food waste management have attracted increasing interest within processing industries [2]. Indeed, international authorities are encouraging the sustainable consumption and modification of production patterns by recommending the “Sustainable Development Goals” (SDGs) [3]. These are focused mainly on: (i) the reduction of food waste; (ii) the development of alternative technologies; and (iii) changing the linear food model (single-use product: produce, consume, and dispose) in favor of new strategies, i.e., the circular economy, that are more efficient and sustainable [2,4,5].

In this context, grapes (*Vitis* spp., Vitaceae) are one of the largest fruit crops produced worldwide. Every year, just over 39.6 million tons (57% of the total crop) are cultivated and processed for the wine industry [6,7]. The genus *Vitis* has more than 70 species; however, the most widely planted species include *Vitis vinifera*, *Vitis labrusca*, *Vitis mustangensis*; *Vitis riparia*, *Vitis rotundifolia*, *Vitis rupestris*, *Vitis aestivalis*, *Vitis arizonica*, and *Vitis berlandieri* [8]. Notwithstanding, the selection of suitable grape varieties to make wines of high quality is a continuous work. The varieties of greatest economic importance to wineries are Cabernet Sauvignon, Sauvignon Blanc, Cabernet Franc, Merlot, Sultana, Grenache, Tempranillo, Riesling, Chardonnay, Syrah, Zinfandel, Touriga Nacional, and Touriga Franca [9]. Despite being one of the highest agro-industrial activities, wineries comprise the sector responsible for producing the greatest amount of industrial waste and by-products (i.e., pruning wood, grape pomace, and lees) [10]. Grape pomace (GP) is the main solid by-product generated through winemaking. It comprises skins, seeds, stems, and disrupted cells from the grape pulp, and it represents ~20–25% of the total weight of processed grapes [11]. Recently, pressure has risen within the wine industry to implement plans for the feasible green disposal or valorization of GP [7].

The valorization of GP for a range of products of industrial interest can be correlated to its complex chemical composition, which is influenced by grape variety, environmental factors, viticulture practices, and the winemaking process [12]. Notably, GP is a valuable source of nutritional and bioactive compounds, including carbohydrates (~12–40%), proteins (~4–15%), lipids (~2–14%), fibers (~17–88%), vitamins and minerals (~2–7%), and polyphenols (~0.2–9%), among others [2,13]. Due to this compositional profile, GP could be used in the food industry as an alternative source of macro/micronutrients for the manufacturing of new functional foods, which can impact human nutrition and health [10].

Particularly, the most relevant GP-derived compounds are polyphenols, e.g., anthocyanins, procyanidins, stilbenes, flavan-3-ols, flavonols and glycosides, and phenolic acids, among others, which can provide several health-promoting effects, as they are recognized as agents with prominent antioxidant, antitumor, anti-aging, and antimicrobial properties [11,14,15,16,17,18,19]. For instance, anthocyanins have demonstrated inhibition of low-density lipoprotein oxidation, which is related to their potential benign effect on cardiovascular diseases [20]. In this regard, after the COVID-19 pandemic, the food industry has encountered relevant challenges in offering functional products with a greater focus on preventative nutrition approaches [21]. Thus, scientists and technologists have focused on searching for new polyphenolic sources to be incorporated into foods [22]. The increase in the number and type of food products containing polyphenols is expected to expand opportunities for the commercial use of grape by-products and thereby increase their economic value [23].

Although GP is a well-characterized food waste, there is a gap in its implementation into food product development and subsequent application in the market. Therefore, the aim of this work was to summarize the current knowledge on the bioactivity and health-promoting effects of polyphenolic-rich extracts from GP samples. Advances in GP incorporation into food formulations (enhancement of physicochemical, sensory, and nutritional quality) and information supporting the intellectual property related to GP’s potential application in the food industry are also discussed.

## 2. Materials and Methods

A comprehensive, structured literature review was performed using the following electronic databases: PubMed, Scopus, Science Direct, Web of Science, and Google Scholar. The following terms or keywords were used to generate a search: “grape pomace wine industry”, “grape pomace composition”, “grape pomace functional properties”, “grape pomace food enriched”, “grape pomace industrial application”, “grape pomace health benefits”. The information contained in the titles and abstracts of articles and patents was screened for the selection of relevant publications in the preparation of this review. For the patent search, we used the keywords “Grape pomace” and “Food industry” to search the World Intellectual Property Organization (WIPOPATENTSCOPE) database and Google Patents.

## 3. Bioactive Polyphenols from Grape Pomace

GP is a valuable source of bioactive compounds with several potential applications, mainly in the food-processing and pharmaceutical industries. Particularly, its recent valorization has focused on the development of new isolation, purification, and recovery procedures to improve its polyphenol profile [23]. The extraction of compounds with green solvents (environmentally friendly solvents with ease of availability from natural and renewable sources, low or negligible toxicity, and biodegradability), such as water, CO_2_, ethanol, and glycerol has demonstrated the richness of GP in bioactive polyphenols, including flavonols (e.g., quercetin, myricetin, kaempferol and their glycosides), anthocyanins (malvidin 3-*O*-glucoside), and phenolic acids (e.g., caffeic acid, *p*-coumaric, etc.). In general, GP polyphenols are classified into three main classes, depending on the number of aromatic rings (attached to hydroxyl groups) present in their structure: (1) phenolic acids (hydroxybenzoic and hydroxycinnamic acids); (2) flavonoids (anthoxanthins, anthocyanins, leucoxanthins, and flavonoidal alkaloids); and (3) nonflavonoids (stilbenes, tannins, coumarins, and neolignans) [24]. A detailed description of the classification is provided by Chakka and Babu [24]. Despite factors such as soil conditions, temperature, cultivation method, grape variety, and geographical origin can strongly impact the chemical composition of GP; the optimization of traditional extraction techniques has assured the concentration and potential activities (e.g., antioxidant, antimicrobial, anti-inflammatory, etc.) of polyphenols [25].

The potential benefits of grape by-products provided by bioactive compounds can also be explored when considering their addition in industrial formulations. For instance, the incorporation of grape flour (Isabella grape), made from by-products, showed interesting results, highlighting higher antioxidant activity (under simulated gastrointestinal digestion), better sensory acceptance, and higher survival of probiotics in an enriched yogurt formulation [26]. The integration of bioactive ingredients into novel food products provides to consumers a new alternative to boost the daily diet with benefits beyond those of nutritional properties. Furthermore, the bioactive richness of GP can be used in the pharmaceutical field which implements antioxidant components into cosmetic formulations for skin care. Some studies have previously elucidated that phenolic compounds and flavonoids play an important role as bioactive components, with the capacity to provide oxidative protection against UV radiation damage. For example, García-Bores et al. [27] reported that the application of *Lippia graveolens* Kunth extracts rich in phenolic compounds decreased the number of lesions seen in laboratory rats exposed to UV damage. Thus, the integration of antioxidant components obtained from GP could be explored to develop new natural ingredients with cosmetic potential and provide an alternative for a full-use model (by-product exploitation). Furthermore, the development of new carrier systems, such as nano-encapsulates and phospholipid vesicles, enhances the number of alternatives for bioactive applications in the industry. The incorporation of bioactive extract components from winemaking by-products (Carignano cultivar), such as quercetin and catechin isomers, among others, comprises a potential component for developing a conventional formulation of cosmetic products, such as creams or ointments, with the capacity to provide antioxidative protection [28].

On this basis, the incorporation of grape by-products into a circular economy could be explored through the exploitation of bioactive constituents such as phenolic acids, tannins, and anthocyanins [29]. Other compounds that have been identified in GP extract are hydroxybenzoic acids, hydroxycinnamic acids, flavonols, flavan-3-ols, stilbenes, isoflavonoids, flavanones, chalcones, hydrolyzable tannins, esters, alcohols, terpenoids, carbonil compounds, furanic compounds, lactones, volatile phenols, and other polar compounds [5]. Thus, due to its active ingredients, GP extract could exhibit antioxidant, anti-carcinogenic, hypotensive, lipid-lowering, and antimicrobial activities [30]. Therefore, winemaking by-products can be applied in the food industry as fibers, polyphenolic extracts, and grape seed oil, or simply combined to produce new foods [31]. Additionally, the exploration of grape by-products as a source of bioactive components has suggested the possibility of developing a new bioactive packaging, as reported by Reinaldo et al. [32].

## 4. Health-Promoting Effects of Grape Pomace

### 4.1. Anti-Cancer Activity

Cancer remains one of the most progressive and harmful diseases worldwide despite considerable progress in basic research and clinical studies [33]. In this sense, the most common treatment for this chronic degenerative disease includes chemotherapy, radiotherapy, and surgical intervention. Nonetheless, recently, authors have demonstrated the anticancer properties of GP polyphenols, describing their bioactivity in different cancer cells. For instance, Balea et al. [33] evaluated the antiproliferative effects of fresh and fermented GP extracts of two *Vitis vinifera* L. varieties, Fetească neagră and Pinot noir. According to the authors, these samples were able to inhibit the proliferation of human lung carcinoma (A549), human breast adenocarcinoma (MDA-MB-231), and murine melanoma B164A5 cell lines at 1 mg L^−1^ after 48 h of incubation, which was correlated to the polyphenols identified in these extracts (e.g., isoquercitrin and quercetrol). These findings were further demonstrated according to a recent study by Spissu et el. [34], who found that polyphenolic-rich extracts of GP (Syrah and Chardonnay varieties) were able to reduce the viability of B16F10 metastatic melanoma cancer cells from 25 to 50% vs. the control, and this effect was dependent on the dose, treatment time, and extract origin. In fact, it has been proven that these antiproliferative effects of in vitro gastrointestinal digested GP extract (Aglianico cultivar) have a great capacity to affect the proliferation of HT29 and Sw480 human colon cancer cell lines [35]. Since gastrointestinal digestion is one of the most important factors involved in the bioaccessibility and functionality of polyphenolic compounds, this study demonstrates the anticancer properties of GP. Furthermore, the solid-state fermentation biotransformation process could be an alternative method to increase and improve the antiproliferative activity of GP against other cancer cell lines, which could be related to the increase and change in the phenolic compounds observed after this bioprocess [18,19]. Thus, GP polyphenols could be considered as natural compounds with antiproliferative properties; however, more studies are needed to demonstrate its anti-cancer properties using additional in vivo systems.

### 4.2. Anti-Cardiovascular Activity

According to Muñoz-Bernal et al. [11], the cardioprotective effect of GP is related to its ability to prevent platelet aggregation by modifying the lipid profile and promoting vasorelaxation. In this line, some studies have been conducted to evaluate the anti-platelet activity of different GP extracts in both healthy volunteers, and patients under stable pharmacological treatment. The first study regarding the use of different GP extracts against platelet-activating factor (PAF) was reported by Choleva et al. [15], who also evaluated other key factors involved in the platelet-aggregation process such as the inhibition of adenosine diphosphate (ADP) and thrombin receptor-activating peptide (TRAP). As reported by the authors, the anti-platelet activity was not grape-variety-dependent, but it did strongly depend on the extraction solvent used for extract recovery. Thus, it was reported that ethanoic extracts were most effective as anti-aggregation agents since their IC50 values were 162.1, 181.2, and 156.3 μg of extract against PAF, ADP, and TRAP, respectively (for samples from healthy volunteers). In addition, it was suggested that this functional property could be more related to the presence of specific phenolic compounds (e.g., malvidin-3-*O*-glucoside, catechin, epicatechin, and quercetin) and other micro-constituents (i.e., fatty acids) than the total polyphenolic content in the GP extracts. These findings were later confirmed by Muñoz-Bernal et al. [16], who indicated that monomeric and oligomeric flavan-3-ols compounds present in the GP from Petit-Verdot-variety grapes could alter the platelet aggregation mechanism, suggesting that this functional property strongly depends on the polyphenolic profile and not on the total polyphenolic content of the GP extracts. However, the synergism and addition effects of polyphenols on the anti-aggregation process should be addressed in the future.

### 4.3. Anti-Diabetic Activity

Since diabetes affects a wide number of people around the word, the study of plant materials containing bioactive compounds and other phytochemicals capable of inhibiting the action of some digestive enzymes involved in the degradation of carbohydrates (α-amylase and α-glucosidase) seems to be a viable approach in controlling this disease. In this frame, there are studies which were conducted with this particular purpose, achieved through different methods of analysis (in silico, in vitro, and in vivo) [36,37,38,39]. It has been reported that some phenolic compounds present in GP, such as peonidin-3-*O*-acetylglucoside, quercetin-3-*O*-glucuronide, isorhametin-3-*O*-glucoside, and catechin, are able to block the active site and inhibit both salivary α-amylase and porcine pancreatic α-amylase enzymes by a competitive effect [39]. However, other enzyme inhibition mechanisms have been recognized as affecting α-amylase activity through the formation of starch–flavonoid complexes during cooking and in the stomach by disturbing enzyme–substrate interaction [40]. Furthermore, the role of other inhibitor agents (e.g., dietary fiber) and more detailed information on this field can be reviewed by readers in a very recent review published by Cisneros-Yupanqui et al. [40], which was not included in this paper to avoid the duplication of information. Despite the published information available, it should be considered that diabetes is a multifactorial disease and more in-depth studies should be carried out to understand the role of the compounds present in GP in its possible control.

### 4.4. Anti-Hyperlipidemic Activity

The blood lipid profile and the presence of white adipose tissue play an important role in the risk of developing cardio- and cerebrovascular diseases [41]. In this sense, GP has been studied to evaluate its anti-hyperlipidemic properties using in vivo models. For example, it has been reported that a diet supplemented with GP improved the lipid profile in the plasma of male Golden Syrian hamsters [41]. In other studies, it was observed that diets supplemented with GP supplied to young rats can reduce total blood triglycerides (TG) and very-low-density lipoprotein (VLDL) levels. On the other hand, high-density lipoprotein (HDL) levels were generally not significantly increased and low-density lipoprotein (LDL) and total cholesterol (TC) levels increased, but these effects depended on the content of GP in the diet and, in some cases, on the feeding time [42,43]. In contrast, Ishimoto et al. [41] reported a decrease in TC values when the diet of male Golden Syrian hamsters was supplemented with GP. Considering that high levels of TG and TC can cause an increased risk of cardiovascular diseases, these results show the potential beneficial effects of GP as a hyperlipidemic agent. These beneficial effects on the lipid profile could be attributed to the presence of high amounts of specific polyphenolic compounds such as catechin, epicatechin, trans-resveratrol, and procyanidins, among others, which are able to affect the lipid profile via different mechanisms, including: (1) increasing fatty acid metabolism through up-regulating carnitine palmitoyltransferase-1A, apolipoprotein-A5 enhancer-binding protein b, and phospholipid transfer protein; (2) inhibiting the plasma secretion of VLDL to reduce lipids in plasma; (3) reducing the absorption of dietary fat in the intestine, and (4) increasing β-oxidation and reducing the secretion of apolipoprotein B [41]. Furthermore, dietary fibers in GP can also play an important role in reducing the intestinal absorption of TG and interfering with the activity of LDL, increasing hepatic receptors, and reducing plasma TG levels [41]. Despite this, relevant information is too limited to establish strong bases regarding the impact of GP as a hyperlipidemic agent, which provides an opportunity for innovative studies in this field.

## 5. Grape Pomace for the Technological and Functional Enhancement of Food

Diet is crucial for human health and its poor implementation can trigger a large number of diseases, some afflicting society in recent years with the incorporation of processed foods and an overall decrease in healthy products [19]. According to Guo et al. [44], poor diet is responsible for one-fifth of the global disease burden; additionally, the spread of COVID-19 during the past number of years raised consumer consideration for healthy products beyond nutritional properties. Thus, at present, market-oriented products concentrate on the performance of essential components; consequently, quantitative research on their main biological activities is important [30]. This section summarizes the most relevant functional properties observed in food formulations when supplemented with polyphenol-rich GP extracts or whole GP (Table 1).

### 5.1. Antioxidant Activity

The addition of GP powder substantially raises the content of bioavailable free polyphenols that can be deficient in other food matrices. Accordingly, GP powder can aid in the formulation of nourishing ready-to-eat products [51]. On the other hand, phenolic compounds are heat-sensitive substances. Consequently, regarding grape pomace, temperatures over 60 °C during pre-treatment and storage could reduce its bioactive compounds [62]. In this sense, other methods, such as lyophilization, could be used to dry GP, have proven to be a good alternative to preserve up to 22% of the available phenolic compounds while also avoiding the use of high temperatures and other undesirable reactions. This is a suitable method for heat-sensitive pigments, but its high associated costs should also be considered [63].

Overall, research conducted on food products enriched with GP has demonstrated a beneficial effect concerning functional properties compared to regular products. For instance, GP-enriched biscuits achieved higher TPC values than the control wheat biscuits did, and they also showed higher antioxidant activity against DPPH^•^ (2,2-diphenyl-1-picrylhydrazyl) and ABTS^•+^ (2,2′-azinobis-(3-ethylbenzothiazoline-6-sulfonate)) radicals [14]. The same effect was observed when aqueous extracts of GP were used to replace water in the formulation of bread [48]. In the same way, considering that wheat flour has a low bioavailability of free polyphenols, cakes were enriched with GP powder and demonstrated higher ash, lipid, protein, anthocyanin, polyphenol, and dietary fiber contents than the control cakes [51]. Moreover, in recent studies, a higher ferric-ion-reducing antioxidant power was registered in bread enriched with 5 and 10% GP compared to traditional bread, even after in vitro digestion [50,64]. According to Gerardi et al. [31], the combination of grape skins and black tea showed a larger ratio of TPC to antioxidant activity than regular black tea. In another study, the addition of 3.5% GP was able to significantly increase the polyphenol content and antioxidant activity of milk chocolate [52]. Since the antioxidant activity of GP polyphenols is well-known and established, high polyphenolic compounds could be related to heightened responses in this functional property.

Thus, the revised information supports the possibility of introducing bioactive compounds from GP into food matrices with the aim of improving desirable characteristics or increasing shelf life. Recently, some researchers have studied the abovementioned integration. As an example, an evaluation of hamburger meat storage under cold temperatures and GP microencapsulation showed a better oxidation stability than the control, with remarkable potential for use as a natural antioxidant in the meat industry [65]. However, the diverse composition of different food matrices leads to new research areas for clarification of the possible mechanisms and compatibility.

### 5.2. Antimicrobial Activity

GP polyphenols have demonstrated an inhibitory effect on the activity of various enzymes and proteins, including bacterial constituents. For example, grape tannins restrain microbial enzymes such as peroxidase, pectinases, lactase, cellulases, and xylanases due to the change in their tertiary structure by non-covalent binging of phenolics to proteins. In the same way, grape flavonoids form structures with proteins which could change the microbial bonding. In addition, GP polyphenols have been proven to influence the integration of Gram-positive bacterial cell walls and to degenerate Gram-negative bacterial outer membranes [66]. In their study, Ghendov-Mosanu et al. [67] found that grape marc extract had an evident bactericidal activity against Gram-positive bacteria, such as *Bacillus subtilis* (ATCC 6633) and *Staphylococcus aureus* (ATCC 25923), in addition to some antibacterial activity against *Escherichia coli* (ATCC 25922). This effect was attributed to polyphenols, which are able to destabilize and change the permeability of the cytoplasmatic membrane and by inhibiting the synthesis of nucleic acids in Gram-negative and Gram-positive bacteria.

The application of GP in the food industry could be desirable, since it involves fewer hazardous effects compared to regularly used synthetic antioxidant and antimicrobial compounds [5]. Saurabh et al. [68] fabricated guar-gum-based films enriched with GP extract (5% *w*/*w*). As a result, the biofilms demonstrated significant antimicrobial activity against *E. coli*, *S. aureus*, *Bacillus cereus*, and *Salmonella Typhimurium*. Furthermore, an improved shelf life of up to 12 days, compared to 8 days for regular films, was demonstrated. Further applications can be found in drug production, accomplishing “medicine and food homology” in food products with grape seed extract that have no toxic effects as a natural food-grade material [30]. In another study, GP extract (GPE) was encapsulated in alginate nanoparticles (Alg-GPE NPs) and chitosan nanoparticles (CS-GPE NPs) through ionic gelation to provide protection from gastrointestinal fluids. Encapsulation in Alg-GPE NPs supported a 3-log decrease in methicillin-susceptible *S. aureus* (MSSA), a 2-log decrease in *Listeria monocytogenes*, *P. aeruginosa*, and *Salmonella enteritidis*, and a 1-log decrease in *E. coli* [69]. The encapsulation of GPE in CS-GPE NPs promoted a 5-log reduction in MSSA, a 3-log reduction in *L. monocytogenes* and *P. aeruginosa*, and a 1-log reduction against *E. coli* and *S. enteritidis*. *Candida albicans* was the most sensitive strain, showing a 6-log reduction in viable cell numbers with both types of NPs [69]. Functional packaging represents a novel research area for natural bioactive compounds beyond a direct integration into food products. A few research projects have studied the potential of GP compounds as a packaging material. A recent study showed the development of a bactericidal isotactic polypropylene (PP) using GP extracts with positive results such as lower water vapor permeability and antimicrobial activity against pathogenic bacterial strains (*E. coli* and *B. subtilis*) [70]. Thus, functional packaging is another research area in which GP compounds can be incorporated, serving as possible shelf-life boosters.

### 5.3. Prebiotic Activity

Prebiotics are defined as “substrates selectively utilized by host microorganisms conferring a health benefit” [71]. Despite the fact that well-established prebiotics are carbohydrate-based (e.g., inulin, fructo-oligosaccharides, galacto-oligosaccharides, and xylo-oligosaccharides), other non-carbohydrate substrates (e.g., phytochemicals, proteins/peptides, and polyunsaturated fatty acids) have also been proposed as prebiotic candidates [71,72]. Recent evidence has demonstrated that phytochemicals, such as polyphenols, have the ability to stimulate the growth of probiotic microorganisms and modulate the gut microbiota in terms of composition and functionality [73]. However, it is worth noting that the prebiotic effect of polyphenols can be influenced by a range of factors, including the food source, chemical structure, and inter-individual differences, etc., [74].

In this sense, Campanella et al. [75] used GP obtained from *V. vinifera* cv. Negroamaro for the growth of the probiotics *Lactiplantibacillus plantarum* 12A and PU1, *Lacticaseibacillus paracasei* 14A, and *Bifidobacterium breve* 15A. Results showed that GP favored bacterial growth by increasing cell density (>9.0 CFU g^−1^). An enhancement of survival under simulated gastrointestinal conditions (1 log cycle compared to the initial cell density) was also observed. Similarly, Dos Santos et al. [56] assessed the effect of GP from *V. vinifera* cv. Pinot Noir on the viability of *Lactobacillus acidophilus* LA-5 and *Lacticaseibacillus rhamnosus* HN001 when added to fermented goat milk. A protective effect of GP on bacterial viability was observed; after 14 days of storage, both strains maintained their populations by up to 7-log CFU mL^−1^, fulfilling the daily intake recommended for a probiotic product. In an interesting study, Chacar et al. [76] evaluated the impact of long-term GP intake (mixture of *V. vinifera* cv. Cabernet Sauvignon, Marselan, and Syrah) on rat gut microbiota. After 14 months of treatment, the ability of GP to selectively modulate the gut microbiome to a healthier phenotype was demonstrated. There was an increase in the relative abundance of Bifidobacterium, Lactobacillus, and Bacteroides, as well as a decrease in Clostridium. More recently, Araújo-Silva et al. [77] evaluated the effect of a goat yogurt formulated with GP from *V. labrusca* cv. Isabel on the growth and metabolism of *Lactobacillus acidophilus* La-05, *Lacticaseibacillus casei* LAFTI L-26, and *Bifidobacterium animalis* subsp. *lactis* Bb-12. Results showed that GP stimulated the growth of bacteria during 48 h of culture, with viable counts of >7 log CFU mL^−1^. Furthermore, it improved bacterial metabolic activity as pH values were reduced (<5) and organic acid production increased (lactic acid = 5–8 mg mL^−1^, acetic acid = 2–4 mg mL^−1^, citric acid = <1 mg mL^−1^, and succinic acid = 0.4–1 mg mL^−1^) in yogurt.

### 5.4. Physicochemical and Sensorial Properties

GP has been used in the manufacturing process of beverages and baked and non-baked products to improve their nutritional value as well as to enhance physicochemical properties and sensorial characteristics. Table 2 presents sensory attributes and key information for the design of new edible products enriched with GP. Some distinctive attributes of GP are increases in acidity, bitterness, and purple color.

For instance, it was shown that 5% GP added into a bread formulation led to a higher acidity and less sweet flavor than regular bread [50] as well as an increased bitter taste in enriched cookies and bread [53,78]. Therefore, it is important to evaluate the food matrix used in the design of the product to obtain a tasteful and balanced flavor. An example of an exceptional blend is GP-enriched chocolate, which received a high score, perhaps because the fruity taste was noted positively, and the darker color did not affect the descriptive evaluation [52]. The performance of different particle sizes has also been studied. GP flour-based pasta (15% *w*/*w*, particle size ≤500 μm) was compared with pasta enriched with GP (15% *w*/*w*, particle size ≤125 μm) and pasta enriched with GP (15% *w*/*w*, particle size ≤125 μm) and 0.6% (*w*/*w*) transglutaminase. The GP with a smaller particle size and transglutaminase recorded an acceptable value of overall quality, higher sensory scores for elasticity and firmness and demonstrated less bulkiness and adhesiveness [59]. Raw materials with a high-fiber content appear to be unacceptable in products such as chocolate, while muesli bars or bread remain acceptable due to the original texture of the product [52].

On the other hand, although antinutritional factors such as biogenic amines (BA) have been found in different non-fermented GP samples, none of the flour samples that were analyzed exceeded the established limits, which could be indicative of a positive effect for food applications in terms of safety [79]. Moreover, it has been reported that GSE is able to reduce the total content of BAs (i.e., putrescine, spermine, spermidine) in tarhana (a cereal-based fermented food) samples when levels over 8 g kg^−1^ were used, with those results depending on the storage period [61]. A similar effect was further observed by Akan and Özdestan-Ocak [60], who reported that GSE suppressed the formation of putrescine at levels of 8–16 g kg^−1^ due to its antioxidant content. Other studies have demonstrated similar effects in total *N*-nitrosamines content when evaluated using western-style smoked sausage [80] and in reducing the formation of the heterocyclic aromatic amines that can be present in heated protein-rich foods such as beef and chicken meatballs [81]. However, special attention should be paid to specific amine-type compounds, such as cadaverine and *N*-nitrosopiperidine, *N*-nitrosodiphenylamine, as their formation has been associated with the addition of GSE into food product formulations [60,80]. Thus, this aspect creates the opportunity to develop novel investigations in this field.

Similarly, Table 2 also shows the chromatic parameters of food products enriched with different amounts of GP. For low-degree additions, color perception could have biased the flavor and acceptability of the product in a favorable way [76]. In a study of functional biscuits, the addition of 2% GP gave a stronger orange-brown tone, which also made it perceivable at a sensory level, but without evident defects [14]. On the other hand, pasta enriched with GP (particle size ≤500 μm) had a low overall quality due to the low resistance and unpleasant dark purple color [59]. Further, minor additions tend to be more visually attractive to consumers and the effect of dark colors depends on the perception of evaluators. Additionally, in some products, such as bread, muffins, cakes, and chocolate, it was detected that the enriched product had a higher overall acceptability than the control product. Therefore, it has been proven that, depending on the conditions of the ingredients, formulations, and processes, it is possible to create savory and nutritional products enriched with GP [51,52,82,83].

Finally, since GP has mainly been used in baked products, Table 3 presents the rheological characteristics evaluated in bread. The vegetative structure used in the product affects its rheological characteristics; for instance, when GP seeds are used, the firmness tends to be higher [49,50,83]. Additionally, grape seed flour grants a preferably neutral taste, and it can result in a sandy texture due to the presence of flavan-3-ols. A possible solution to enhance the texture could be the use of purified extracts [49,52]. In a study where authors compared different vegetable by-products, the crusts of bread enriched with GP extract had the highest firmness and crunchiness [48]. The firmness of the bread could be explained by the unfolding of the gluten network, resulting in acidic conditions and an increase in positive charges. This reduced the extension of the dough, the volume, and the specific volume. The addition of GP could reduce the strength of the gluten network due to poor gas-retention ability; consequently, as the amount of added GP increased, the firmness also increased [50]. The synergy of fiber and gluten augmented the mixing tolerance index (MTI), and the fortified biscuits were less thick, possibly due to the dilution of the gluten protein [45].

## 6. Intellectual Property in Applications of Grape Pomace

In order to visualize the technological development and novel trends surrounding the industrial application of GP, an analysis was implemented through the world intellectual property database “WIPO-PATENTSCOPE” (accessed on 22 April 2023). In total, 27 patents were registered in 2023. Thus, patents wherein the application of GP encompasses a direct main component or an auxiliary in the development process are described in Table 4. The first patents involve the development of new bioactive food products. The patent RU0002783431 refers to a drink with GP extract as a sugar-containing liquid enriched with *Zygosaccharomyces kombuchaensis* and *Gluconacetobacter xylinus*; the obtained beverage demonstrates biological activity, with improved taste qualities. Similarly, the patent RU0002775316 encompasses the production of functional tablets by the incorporation of apple puree, lactic acid, and grape powder, among other constituents. In addition, the incorporation of GP can be related to an alternative use and not for its bioactive properties; for example, the patent CN114601152 describes an *Actinidia arguta* product rich in dietary fibers, during the processing of which GP and other fruit by-products are applied. Similarly, it has been reported that the application of GP in *Lentinus edodes* production media (patent CN114747424) increases anthocyanin levels in mushrooms.

The following patents are related to the production of an initial material that can be applied in the development of other food products. The patent CN114951239 discloses a method for preparing a raw material by recovering GP; the invention provides a stable raw material (low amounts of high-value components loss). The second patent is related to animal feed (patent CN114568584) via the preparation of grape-branch-coated silage that retains the biological properties of grape branches with complete nutritional effects. As the most recent patents, patents US20230022145 and US20230055369, disclose methods of GP preparation and its applicability in beverage production and in the application of GP in generating protein–polyphenol complexes, respectively. The intellectual property related to GP applications shows that the range of applications in the industrial sector is wide and technological development is focused on taking advantage of the beneficial properties of the residual material. This facilitates the transformation of agro-industrial by-products into a source of bioactive components, with possible impacts on health and food technology.
foods-13-00580-t002_Table 2Table 2Chromatic and/or sensory characteristics of food products enriched with grape pomace.Food Matrix ModelGrape VarietyVegetative  StructureGrape Pomace  ProcessingGrape Pomace  Extract  Composition (%)Chromatic ParametersSensory EvaluationReferencea*b*L*BiscuitNRSkinsLyophilization106.5221.9565.84Sweeter taste and reduced  thickness[45]*Vitis vinifera* L. cv. SangioveseWhole grape pomaceLyophilization and hydroalcoholic  extraction28.8 ^a^9.7424.2241.14Higher score for all the sensory attributes, excluding friability[14]Bread*Vitis vinifera* L. cv. MerlotSeedsNR510.216.554.9Higher overall acceptability[82]*Vitis vinifera* L. cv. EmirWhole grape pomaceNR56.4815.6858.47Sensory characteristics not statistically significant compared to the control[46]NRNRMicrowave-assisted aqueous extraction34.5 ^a^7.514.9927.88Sweeter, sourer, and had the best crust firmness and crunchiness[48]*Vitis vinifera* L. cv. ZelenSkins and  seedsConvection drying63.9813.1151.44Reduced sensations of sandiness, toughness, and taste. Sour smell was perceived[49]NRDefatted seedsMechanical  degreasing5NENENESimilar characteristics to those of the control[78]*Vitis vinifera* L. cv. CorvinaSkinsVacuum drying54.2517.4548.34Lower scores for acidity, astringency, and wine smell[50]Cake*Vitis vinifera* L. cv. Muscat HamburgWhole grape pomaceConvection drying4NENENEHigher appearance, taste, and odor[51]ChocolateNRWhole grape pomaceConvection drying7.50NENENEHigher color, taste, texture, and overall popularity[52]Cookie*Vitis vinifera* L. cv. MoscatSkin and seedsConvection drying6NENENEHarder texture and a more bitter taste compared to the control[53]Meat*Vitis vinifera* L. cv. MonastrellSkinsAlcoholic extraction by high or low  instantaneous  pressure0.06 ^a^9.582.9141.93NE[84]Milk*Vitis vinifera* L. cv.  Pinot noirWhole grape pomaceNR2 ^a^NENENEBetter flavor, color, and overall acceptability[56]Muffin*Vitis vinifera* L. cv. CabernetSkins and seedsNR3NENENEHigher sensorial attributes and firmness score. An interesting purple color[83]PastaNRSkins, seeds, and stalksConvection drying15NENENEBetter elasticity, firmness, and  acceptable overall quality[59]NR = not reported; NE = not evaluated; ^a^ = GP extract.
foods-13-00580-t003_Table 3Table 3Rheological characteristics evaluated in bread formulations enriched with grape pomace.Vegetative  StructureGrape Pomace Extract Composition (%)Rheological ParametersReferenceWater ActivityMoisture Content (%)Firmness (N)Volume (cm^3^)Specific Volume  (cm^3^ g^−1^)Skins, seeds, leaves, and stems50.92NRNR14633.17[46]Skins and seeds50.9528.82NR620NR[47]Skins, seeds, leaves, and stems34.5 ^a^NR42.16NR23923.05[48]Skins and seeds6NRNR125010802.72[49]Skins and seeds3NRNR786.26NRNR[83]Skins50.9741.4621.816913.59[59]NR = not reported; ^a^ = GP extract.
foods-13-00580-t004_Table 4Table 4Current patents in WIPO-PATENTSCOPE database related with industrial application of grape pomace.Patent NumberTitleMain CoreScopePublication DateCountryRU0002783431Method for preparation of  a drink with biological  activityThe invention belongs to the food and processing industry to produce soft drinks with biological activity. The method for obtaining the drink involves the use of grape pomace extract as a sugar-containing liquid. Cultures of yeast *Zygosaccharomyces Kombuchaensis* and *Gluconacetobacter xylinus* are introduced in fermentation. The invention makes it possible to obtain a drink with biological activity, with improved taste qualities.Food  technology14 November 2022Russian  FederationRU0002775316Method for production of pastilles with functional propertiesThe invention belongs to the food industry and can be used to make tablets with functional properties. Pastilla mass preparation consists of apple puree, whipped with sugar, in addition to cooked and cooled agar sugar molasses syrup, lactic acid, and a functional additive consisting of grape powder obtained from grape pomace. The invention provides a tablet with functional properties, the consumption of which normalizes the nutritional status of the functional ingredients of food.Food  technology29 June 2022Russian  FederationCN114601152*Actinidia arguta* product rich in dietary fibers and lactic acid bacteria and preparation method of *Actinidia*

*arguta* productThe invention relates to a product of *Actinidia arguta* rich in dietary fiber and lactic acid bacteria, as well as the associated method of preparation. The damage to the functional components of raw materials in processing is reduced, and the final product is rich in dietary fiber and probiotics. The obtained product can also be added to other products as a core material and strengthens gastrointestinal health.Food  technology10 June 2022ChinaCN114747424*Lentinus edodes* culture  medium with anti-aging  effectThe invention provides a *Lentinus edodes* culture medium with an anti-aging effect and aims to solve the poor anti-aging  effect of *L. edodes*. *L. edodes* cultivated by adopting the substrate disclosed by the invention contains relatively high NMN (beta-nicotinamide mononucleotide) content and has a relatively good anti-aging effect.Waste  revaluation15 July 2022ChinaCN114951239Method for preparing derivative raw material by recovering grape pomace in wine-brewing processThe invention discloses a method for preparing a derivative raw material by recovering grape pomace in a wine-brewing process. According to the method, microorganisms in the wine residues are eliminated, the properties of raw material products are stable, and the volatilization and loss of high-value products are avoided. Finally, the problems of traditional raw materials (deterioration and storage difficulties)  are effectively solved.Food  technology30 August 2022ChinaCN114568584Preparation method of grape-branch-coated silageThe invention describes a method of preparing grape-branch-coated silage that retains the biological properties of grape branches. The product comes from a mixture of grape pomace and selenium yeast, so that the functional grape branch silage is endowed with more complete nutritional effects and interesting characteristics such as good palatability and easy  digestion.Waste  revaluation3 June 2022ChinaUS20230022145Grape skin compositions and compounds, and methods of preparation and use thereafterThe patent encompasses methods of preparing grape pomace compositions, and methods of using these compositions and compounds. Particularly encompassed are methods of using grape skins for beverage preparation.Food  technology26 January 2023USAUS20230055369Plant-sourced protein–polyphenol complexesThe patent discloses compositions and methods for one or more plant protein(s) combined with vegetal sources of phenolics (plant waste, e.g., pomace) to generate protein–phenol complexes. This mixture constitutes an admixture that can be added to meat-analogue formulations.Food  technology23 February 2023USA

## 7. Conclusions and Future Perspectives

GP valorization is a step toward food sustainability. Current evidence supports the potential application of GP within the food industry. Particularly, GP polyphenols have been proven to exert a wide range of biological activities (e.g., antioxidant, antimicrobial, prebiotic, anti-proliferative, anti-lipidemic, etc.), which may aid in treating certain diseases or syndromes, including diabetes, hypertension, obesity, aging, cancer, and neurodegenerative diseases, among others. Hence, GP polyphenols could provide functional advantages in the development of novel products. Additionally, the incorporation of GP into food formulations could also offer technological advantages for the industry. The studies summarized in this review have demonstrated that GP can enhance the physicochemical, sensory, and nutritional quality of food products. On the other hand, the referenced intellectual property demonstrates that progress in the application of GP has remained highly niche. The number of granted patents is specific to China, Russia, and the USA, indicating that other countries and future patents should consider new advances in incorporating GP into food.

Despite the promising effects of GP, the development of GP-enriched foods remains one of the most challenging research areas, thus further research is needed regarding this topic. For instance, future studies should focus on the extraction, identification, and characterization of novel polyphenols derived from GP. Likewise, microbial interventions (fermentation by adding starter cultures) can be conducted to improve the nutritional and bioactive characteristics of GP. Alternatively, further studies could optimize proposed formulations to achieve the highest quality and functionality of GP-enriched food. Finally, more pilot-scale studies, along with economic evaluations, are necessary to determine the commercial reliability of manufacturing GP food products. If all these gaps are overcome, advances in GP valorization through a circular economy perspective will allow us to evolve from waste to health.

## Figures and Tables

**Table 1 foods-13-00580-t001:** Phenolic composition of food products enriched with grape pomace.

Food Matrix Model	Grape Variety	Vegetative Structure	Grape Pomace Processing	Grape Pomace Extract Composition (%)	Total Phenolic Content (mg GAE g^−1^)	Phenolic Compounds	Reference
Beverage	*Vitis vinifera* L. cv. Primitivo	Skins	Convection drying	2	0.200 ± 0.17	NR	[31]
Biscuit	White pomace	Skins	Lyophilization	10	2.11 ± 0.07	Gallic acid, tyrosol, ɣ-resorcylic acid, catechin, isovanilic acid, procyanidin B1, *trans*-ferulic acid, *p*-coumaric acid	[45]
	*Vitis vinifera* L. cv. Sangiovese	Whole grape pomace	Lyophilization and hydroalcoholic extraction	28.8 ^a^	0.629 ± 0.31	Delphinidin 3-*O*-β-D-glucoside, cyanidin 3-*O*-β-D-glucoside, petunidin 3-*O*-β-D-glucoside, malvidin 3-*O*-β-D-glucoside, malvidin-3-*O*-(6-*O*-*p*-coumaroyl)-β-D-glucoside	[14]
Bread	*Vitis vinifera* L. cv. Emir	Whole grape pomace	NR	5	0.675 ± 0.07	NR	[46]
	*Vitis vinifera* L. cv. Pinot Noir	Skins and seeds	Convection drying	5	0.7	NR	[47]
	NR	NR	Microwave-assisted aqueous extraction	34.5 ^a^	0.951 ± 0.15	NR	[48]
	*Vitis vinifera* L. cv. Zelen	Skins and seeds	Convection drying	6	2.42 ± 0.41	NR	[49]
	*Vitis vinifera* L. cv. Corvina	Skins	Vacuum drying	5	1.01 ± 0.07	NR	[50]
Cake	*Vitis vinifera* L. cv. Muscat Hamburg	Whole grape pomace	Convection drying	4	0.0941 ± 0.003	Tyrosol, catechin, gallic acid, quercetin, epicatechin derivative, protocatechuic acid, kaempferol, apigenin, catechin derivative	[51]
Chocolate	NR	Whole grape pomace	Convection drying	7.5	0.521 ± 0.11	NR	[52]
Cookie	*Vitis vinifera* L. cv. Muscat	Skin and seeds	Convection drying	6	4.03 ± 0.37	NR	[53]
Fiber-rich extract	*Vitis vinifera* L. cv. Alicante Bouschet	Seeds	Convection drying and hot water- assisted extraction	100 ^b^	0.016	NR	[54]
Dairy product	*Vitis vinifera* L. cv. Timorasso	Whole grape pomace	Convection drying and aqueous extraction	NR	0.060	NR	[55]
	*Vitis vinifera* L. cv. Pinot Noir	Whole grape pomace	Hydroalcoholic extraction	2 ^a^	0.045	NR	[56]
	*Vitis vinifera* L. cv. Barbera	Whole grape pomace	Convection drying, high pressure- and temperature-assisted extraction	3.7	0.08	NR	[57]
Pasta	*Vitis vinifera* L. cv. Gamay	Whole grape pomace	Convection drying	7	0.061	Quercetin, maldivin-3-*O*-glucoside maldivin-3-*O*(6-*O*-*p*-coumaroyl) glucoside, peonidin-3-*O*-glucoside, kaempferol, peonidin-3-*O*(6-*O*-*p*-coumaroyl), petunidin-3-*O*-glucoside, cyanidin-3-*O*-glucoside, delphinidin-3-*O*-glucoside	[58]
	NR	Skins, seeds, and stalks	Convection drying	15	2.11	NR	[59]
Cereal-based fermented food	NR	Seeds	NR	0.8	2.504–3.020 ^c^	NR	[60]
					1.636–1.773 ^d^	NR	[61]

NR = not reported; GAG = gallic acid equivalents; ^a^ = GP extract; ^b^ = grape seed extract; ^c^ = depending on the storage period; ^d^ = depending on the fermentation period.

## Data Availability

Data is contained within the article.

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
