# Peer review of "Grape Pomace—Advances in Its Bioactivity, Health Benefits, and Food Applications"

_foods, 2024, doi:10.3390/foods13040580_

Round 1

Reviewer 1 Report

Comments and Suggestions for Authors

The review article is generally very well written, both in its coverage of material relevant to the topic chosen, and also in the written English. The article focuses on the polyphenolic content of grape pomace and reviews supporting literature on potential health benefits and summarizing past studies on this topic.  The article is well worth for Foods, while some issues around terminology, particularly with regards to phenolic classes, should be undertaken.

1.       Regarding materials and methods; the literature searches involved phrases containing “grape pomace”; were any additional searches made with the phrase “grape marc”, given that is often used as an alternative to “grape pomace”.

2.       Line 103: suggest to change “rutin and kaempferol” to “quercetin, myricetin, kaempferol and their glycosides”. This is because the most common flavonols in grape are often quercetin-3-glucoside and quercetin-3-glucuronide, with myricetin glycosides also found; the aglycones are rare in the grape and appear more during winemaking.

3.       Line 104: suggest to change “cyanidin-3-O-glucoside” to “malvidin-3-O-glucoside”, given that the latter is usually the major monomeric anthocyanin present in red grapes; in some varieties such as Pinot noir, cyanidin-3-O-glucoside can be below detection limits.

4.       Line 120, 134 and elsewhere: the terms “flavonoids” and “tannins” are introduced, but defining these terms and how they include particular phenolic compounds mentioned elsewhere.  For example, most condensed tannins from grapes are oligomers of flavan-3-ols, which belong to the flavonoid family (as do anthocyanins).  A small extra section to introduce the phenolic compound discussed in the article and their associated terminology would be helpful.

5.       Most sections end with comments about the need for more studies to be undertaken, and provide balanced conclusions (e.g. lines 212 and 238).  The final sentence for the anti-cancer section 4.1 offers the conclusion that “…GP polyphenols can be considered as a natural therapy to prevent cancer diseases”.  This conclusion appears too strong and not consistent with the evidence available to date.

6.       Line 201: suggest to replace “some anthocyanins” with “some phenolic compounds”, given that quercetin-3-O-glucuoronide and catechin are not anthocyanins.

Author Response

All your concerns were attended. You can see more details in the attached file.

Reviewer 2 Report

Comments and Suggestions for Authors

1 - Introduction: The review discusses grape cultivation, and at no point in the introduction are the cultivated species mentioned. In the introduction section, the main cultivated species and those of greater economic importance will be addressed.

2 - Introduction: Addressing how global warming can impact grape production.

3 - Anti-Hyperlipidemic Activity: Discussing possible mechanisms of anti-hyperlipidemic activity.

4 - Antimicrobial Activity: Provide a more detailed explanation, outlining the mechanism of action by which grape tannins restrain microbial enzymes such as peroxidase, pectinases, lactase, cellulases, and xylanases due to changes in their tertiary structure.

5 - Antimicrobial Activity: (ATCC 6633) - Explain in more detail.

6 - Remember to cite the individual who described each plant species when mentioning the scientific name for the first time, following the rules of scientific naming.

Comments on the Quality of English Language

Extensive editing of English language required.

Author Response

(The authors gave the same response as above.)
